# Oncologic outcomes of Bacillus Calmette-Guérin therapy in elderly patients with non-muscle-invasive bladder cancer: A meta-analysis

Seyed Mohammad Kazem Aghamir[1], Fatemeh Khatami[1], Hossein Farrokhpour[2], Leonardo Oliveira Reis[3], Mahin Ahmadi Pishkuhi[4], Abdolreza Mohammadi[1]*

1 Urology Research Center, Tehran University of Medical Sciences, Tehran, Iran, 2 Department of Medicine, Tehran University of Medical Sciences, Tehran, Iran, 3 UroScience and Department of Surgery (Urology), School of Medical Sciences, University of Campinas, Unicamp, and Pontifical Catholic University of Campinas, PUC-Campinas, Campinas, São Paulo, Brazil, 4 Pars Advanced and Minimally Invasive Medical Manners Research Center, Pars Hospital, Iran University of Medical Sciences, Tehran, Iran

* ab2rezamohammadi@yahoo.com

**Data Availability Statement:** All relevant data are within the manuscript and its Supporting information files.

## Abstract

### Introduction

There is a challenge on the medical efficacy of intravesical Bacillus Calmette-Guérin (BCG) therapy and the power of the immune system boosting, which can be influenced by the age of the non-muscle-invasive bladder cancer (NMIBC) patients. This meta-analysis evaluates the efficacy of BCG therapy among aged (>70) and younger patients with non-muscle-invasive bladder cancer (NMIBC).

### Methods

The central database of PubMed, Scopus, and Web of Science were queried until August 4, 2021, by using "BCG," "Bladder Cancer," "AGE," and "efficacy" keywords. After excluding duplicated results, titles and abstracts were evaluated by two independent reviewers. The exclusion criteria included non-English studies, conference abstracts, reviews, editorials, letters, and comments. Three main outcomes, disease-free survival (DFS), progression-free survival (PFS), and cancer-specific survival (CSS), were considered. The statistical analysis was performed using STATA (version 14; Stata Corp, College Station, Texas, USA).

### Results

From 1115 found documents, the 24 research articles were recruited in the systematic review, and 10 were the candidate for meta-analysis. The overall estimate of H.R. revealed that BCG therapy in those over age 70 is significantly associated with an improved risk of progression and cancer-specific death in studied patients. However, this association was not statistically significant for DFS (1.04 (95% CI: 0.85,1.26)).

**Funding:** The authors received no specific funding for this work.

**Competing interests:** The authors have declared that no competing interests exist.

**Abbreviations:** APC, Antigen-presenting cells; BCG, Bacillus Calmette-Guerin; CIS, Carcinoma in situ; CSS, Cancer-specific survival; DFS, Disease-free survival; IFNγ, Interferon-gamma; IL-12, Interleukin-12 (IL-12); N.K., Natural Killer cells; NMIBC, Non-muscle-invasive bladder cancer; PFS, Progression-free survival; Th1, lymphocytes T helper cells; TNFα, Tumor necrosis factor Alfa (TNFα); TURBT, Transurethral resection of bladder tumor.

## Conclusion

The BCG maintenance therapy improved CSS and PFS oncological outcomes in elderly patients with NMIBC. BCG therapy did not significantly change the DSF.

## Introduction

Bladder cancer (BC) is the 10th most common cancer worldwide, increasing the risk as people become older [1]. This cancer is more common in men than women (4:1). It is the sixth most common cancer and the ninth leading cause of cancer death in men [2]. The average age of diagnosis at initial presentation is 73 years in the united states [3, 4]. Due to the expanding life expectancy, bladder cancer merits more specific attention [5]. Treatment of non-muscle invasive bladder cancers (Ta, CIS, T1) is based on risk stratification. High-grade bladder cancers (HGTa, HGT1, CIS) have a high possibility of disease recurrence and progression to muscle-invasive tumors [6]. Intravesical instillation of Bacillus Calmette–Guèrin (BCG) into the bladder after transurethral resection of the bladder tumor (TURBT) is the gold standard for conservative treatment for high-grade T1 disease. The most accepted BCG administration protocol is an initial 6-weekly intravesical instillation followed by maintenance therapy of 3-weekly installation in 3,6 and every six months interval for 1–3 years according to the risk category groups. (1 year in intermediate and 1–3 years in high-risk patients) [7]. This protocol is the most cited regimen to reduce tumor recurrence. In many studies, the BCG superiority over intravesical chemotherapy is proven [8–10]. The mechanism of the BCG is triggering and activation of both CD4$^+$ and CD8$^+$ T cells into the bladder wall, recruiting an anticancer consequence intermediated by the communication of antigen-presenting cells (APCs) and lymphocytes T helper cells (Th1). Finally, activation of CD8+ cytotoxic, Natural Killer cells (N.K.) and shedding of inflammatory molecules like Interferon-gamma (IFNγ), Interleukin-12 (IL-12), and tumor necrosis factor Alfa (TNFα). This cascade of inflammatory responses destroys floating tumor cells and subsequently decreases tumor recurrence [11, 12]. The therapeutic response to intravesical BCG is dependent on the response of the innate immune system, influenced by the patient's age [13]. Since the innate and adaptive immune systems are weakened in the elderly patients and ongoing concerns exist regarding the efficacy of BCG therapy in patients >70 years, we designed a study to systematically review and evaluate the effectiveness of the BCG in elderly patients [14].

## Methods

### Search strategy

This systematic review and meta-analysis were carried out following the Preferred Reporting Items for Systematic Reviews and Meta-Analyses (PRISMA) statement. cThe databases, including PubMed, Scopus, Web of Science, were queried until August 4, 2021, utilizing keywords "AGE" AND "BCG" AND "Bladder Tumor" with no filter, including all the results. Additionally, the reference list of identified articles was manually searched to obtain different results. A detailed search strategy regarding each database is demonstrated in the Supplementary Material.

### Study selection

After excluding duplicated results, titles and abstracts were evaluated by two independent reviewers. Studies examining the impact of age on the clinical efficacy of BCG therapy in

patients with non-muscle invasive Bladder Cancer were included for further investigation. Since the age effect was not mentioned directly in the abstract, all possible candidate studies were marked to be thoroughly reviewed. Two independent reviewers performed full-text screening of the selected studies, and discrepancies were solved through mutual discussion. Inclusion criteria were all patients with a histopathological diagnosis of non-muscle-invasive bladder cancer who received a BCG induction and maintenance therapy. We did not exclude recurrent, multiple, or CIS tumors.

Given that age has been included as a potential factor in many studies, to make comparison possible, only studies in which age was defined as categorical with specific cut-off values (70 or 75 years) were included. The negligible difference between these two classifications of ages was ignored. The exclusion criteria included non-English studies, conference abstracts, reviews, editorials, letters, and comments.

## Outcomes, data extraction, and quality assessment

Three main endpoints were evaluated: 1. Disease-free survival (DFS) (recurrence-free survival), 2. Progression-free survival (PFS), 3. Cancer-specific free survival.

Data was then extracted using a pre-designed form to obtain basic study features, study population and design, outcomes, H.R.s and 95% C.I.s, follow-up period, and risk of bias. Two well-trained authors extracted data and were double-checked by a third investigator to validate the accuracy. Statistical calculations were used to indirectly estimate H.R. and related C.I.s where they were not reported.

Two independent authors conducted quality assessments, and discrepancies were followed by discussion. Quality assessment was performed using The Newcastle-Ottawa scale (NOS) to evaluate the risk of bias and the Quality Assessment of Diagnostic Accuracy Studies 2 (QUADAS-2) tool to examine the methodological quality of diagnostic studies.

## Statistical analysis

The statistical analysis was performed using STATA (version 14; Stata Corp, College Station, Texas, USA). A P-value less than 0.05 was defined as statistically significant. To evaluate the impact of age with a 70-year-old cut-off on DFS and PFS, Pooled H.R.s and 95% C.I.s were calculated. In patients older than 70, H.R.s with values higher than 1 demonstrated the poor prognosis of older patients who receive BCG therapy compared to younger patients.

To assess statistical heterogeneity, two tests were conducted: 1. The Higgin's I-square test with $I^2 > 50\%$ indicates heterogeneity, and 2. The Cochrane's Q test with P-value $< 0.05$ states heterogeneity. A random effect was employed in heterogeneity (I2 > 50%). The likelihood for publication bias was assessed using a funnel plot, Begg's and Egger's tests, and Duval and Tweedie's trim-and-fill method.

## Results

### Study selection and characteristics

The strategies and results of study selection are presented in Fig 1. A total of 1115 articles remained after duplication deletion, and finally, 24 related articles were analyzed further. Of 24 articles, ten studies include H.R. for DFS and PFS that finally were the candidate for meta-analysis.

Table 1 represents a summary of findings in the candidate articles. We chose the papers that had an initial six weekly induction therapy and maintenance therapy protocol. The

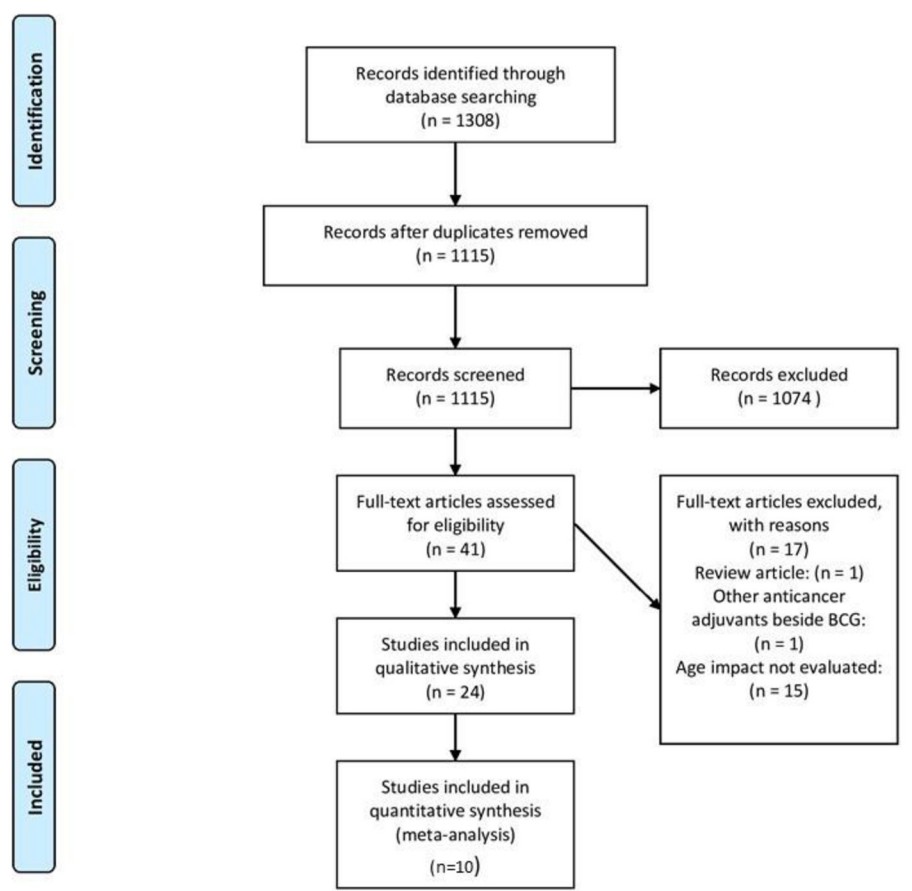

**Fig 1. Flow diagram of study selection for the current meta-analysis.**

patient's age included in the analysis should be mentioned in articles as the categorical range (<70 years vs. ≥70 years).

In eight studies, the central concept that older age unfavorably changes the outcome of patients with NMIBC and advanced age is associated with higher progression rates despite BCG. It is suggested that the care of BC in the elderly population is of growing worry and should be considered in future clinical trials. Advanced age was defined differently from 60 to 80 years in studies. A survey by Milošević et al. shows that the sex and age of patients may have a substantial impact on the course and outcome of NMIBC [14–21]. The NMIBC has malignant and aggressive trends when it happens in males older than 60 years [20].

Six other articles reported contradictory results, which indicated age does not certainly affect recurrence in patients with bladder cancer treated with BCG therapy [22–36]. Moreover, no significant result for age, gender, or response to BCG therapy as predictors of outcome. Patients presenting with primary CIS have a worse outcome than those with secondary CIS [26]. An additional multivariant study by Alvarez-Mu'gica et al. specified that among sex, age, multifocality, tumor size, and concomitant carcinoma in situ, just *PMF-1* methylation makes hazard ratios (H.R.s) for recurrence and progression [30].

Finally, ten studies were included in the meta-analysis, representing 45,336 patients with BC. Of these ten studies, 9 examined H.R. for DFS, 8 examined H.R. for PFS, and 6 reviewed H.R. for CSS [14, 15, 18, 19, 24, 28, 32, 35–37]. Descriptive characteristics are listed in Table 2.

**Table 1. A review of 24 candidate article contents and their conclusion on the impact of age on the BCG therapy efficacy.**

| | Year | First Author | Country | Sample Size | Mean Age (years) | Male | Cancer Type | Treatment Protocol | Follow-up | Conclusions |
|---|---|---|---|---|---|---|---|---|---|---|
| 1 | 1998 | Takashi | Japan | 84 | 65.3+10. | 75% | CIS | Weekly administered for 8 or 10 weeks. No maintenance therapy | 56 months | The response to intravesical BCG therapy may have a role in the reduced host immunocompetence in elderly individuals. |
| 2 | 2004 | Andius | Sweden | 173 | 70+0.7 | 84.90% | CIS | Six weeks of induction instillation and monthly maintenance for 1–2 years were performed routinely | 73 months as mean follow up 72 months (6–154 months) | Recurrence and progression is not dependent on age, number of tumors, number of positive cystoscopies, length of tumor history before BCG, BCG strain. |
| 3 | 2007 | Herr | USA | 805 | 65(24–93) | 76% | Multiple or recurrent high-grade Ta, T1, and carcinoma in situ | Six weekly instillations of BCG Therapy. | Six months and five years | Aging has a measurable but small impact on the overall outcomes of high-risk superficial bladder cancer. |
| 4 | 2008 | Fernandez-Gomez | Spain | 1062 | 66(58–72) | 89.50% | Ta 214 (20.2) / T1 848 (79.8) | Instillation was repeated once weekly for 6 consecutive weeks and thereafter once every 2 weeks, six times more. | 65 months | Age, history of recurrence, high grade, T1 stage, and recurrence at first cystoscopy were independent predictors of progression |
| 5 | 2008 | Takenaka | Japan | 185 | 68.2(39–91) | 83% | CIS | Instillations of 80 mg (total,640 mg) of Tokyo 172 strain BCG in 40 mL of normal saline, starting two to three weeks after diagnostic biopsies. | 37.5 month | The extent of Bacillus Calmette-Guerin (BCG) to treat carcinoma in situ (CIS) might be the only prognostic factor. Disease progression, including extravesical involvement, should be carefully monitored over the long term after BCG therapy. |
| 6 | 2010 | Boorijan | USA | 1021 | 64 | 74% | | Induction (weekly treatments for six weeks) BCG therapy for NMI UC of the bladder | Five years | The outcomes of men and women with high-risk non-muscle-invasive urothelial carcinoma in both age groups (> 50 and <50) treated with BCG are similar. |
| 7 | 2010 | Chade | USA | 476 | 66.7 (13.1) | 82% | Primary or secondary y CIS | The course of 6 weekly intravesical Instillations. | 5.1 years | We found no significance for age, gender, or response to BCG therapy as predictors of outcome. Patients presenting with primary CIS have a worse outcome compared to those with secondary CIS. |
| 8 | 2010 | Kohjimoto | Japan | 491 | 69(22–92) | 86% | NMIBC(TCC) | BCG (Tokyo 172 strain) administered weekly for six consecutive weeks except for | 44.5 months | Older age adversely affected the outcome of patients with NMIBC, which is particularly apparent in patients 80 years or older. Further prospective studies to confirm these findings are warranted. |
| 9 | 2011 | Margel | Canada | 238 | 70 years (range 28–90) | 76% | T1, Ta, CIS | Intravesical BCG (81 mg in 50 mL normal saline) once a week for six continuous weeks. Maintenance BCG (Lamm protocol) was considered. | 38 months | Older age is related to the higher progression rates despite BCG. |
| 10 | 2011 | Yuge | Japan | 447 | - | 82.30% | CIS | - | - | Age does not certainly affect recurrence in patients with bladder cancer treated with BCG therapy. |

*(Continued)*

**Table 1.** (Continued)

| | Year | First Author | Country | Sample Size | Mean Age (years) | Male | Cancer Type | Treatment Protocol | Follow-up | Conclusions |
|---|---|---|---|---|---|---|---|---|---|---|
| 11 | 2013 | Ajili | Tunisia | 112 | - | 90.20% | High-risk NMIBC | Instillations of BCG (BCG Pasteur strain, 75mg in 50mL saline), 3–6 weeks after the last transurethral resection urinary cytology and cystoscopy examination were performed. | 30 months | Aging has no impact on the outcomes of high-risk NMIBC treated by BCG immunotherapy. |
| 12 | 2013 | Dal Moro | Italy | 341 | 63.6 years, 28–88 | 85.40% | Ta-T1 HG and carcinoma in situ | - | 60 months | In patients with primary high-risk BC suitable for BCG treatment, age is not a factor predictive of recurrence or progression of the disease. |
| 13 | 2013 | Alvarez-Múgica | Spain | 108 | 65.6±9.7 | 92.60% | Primary-stage T1HG urothelial carcinoma with complete TUR | Installation 1 (BCG, Connaught strain) was given 14 d after TUR, repeated weekly for 6 consecutive weeks, and thereafter every 2 weeks for six additional instillations within 5–6 months after TUR | 77 months (range: 5–235 months). | Multivariate analyses indicated that among sex, age, focality, tumor size, and concomitant carcinoma in situ, only PMF-1 methylation provided significant hazard ratios (H.R.s) for recurrence and progression. |
| 14 | 2014 | Oddens | Belgium | 546 | | 78.60% | Intermediate- or high-risk Ta T1 (without carcinoma in situ) NMIBC | BCG with or without INH weekly for 6 consecutive weeks starting 7–15 d after TUR. initial six instillations were followed by three weekly instillations at months 3, 6, 12, 18, 24, 30, and 36. | 9.2 years | In intermediate- and high-risk Ta T1 urothelial bladder cancer patients treated with BCG, patients >70 years of age have a worse long-term prognosis. |
| 15 | 2015 | Gontero | Italy | 2451 | 68 years | 82.10% | T1G3(a high-risk subgroup of non–muscle-invasive bladder cancer (NMIBC) | At least an induction the course of BCG as their initial intravesical treatment | 5.2 years | In a subgroup of T1G3 patients with age ≥70-year, tumor size ≥3 cm, and concomitant CIS, the higher risk of progression and thus require aggressive treatment was suggested. |
| 16 | 2015 | Milošević | Serbia | 899 | 61.05 ± 10.52 | 73.40% | NMIBC | Only induction therapy | - | The sex and age of patients may have a significant influence on the course and outcome of NMIBC. The disease has the most malignant and most aggressive behavior when present in males older than 60 years. |
| 17 | 2018 | Hurle | Italy | 185 | 72(66–78) | 77.30% | T1 Highly grade bladder cancer | Induction and maintenance courses (at 3 and 6 months after the induction course and every six months thereafter till 36 months), second induction if tumor recurrence was detected immediately after the first induction course. | 93 months (63–147) | Intravesical BCG appears to be an effective treatment for H.G. pT1 BC Caution should be used in patients aged ≥70 years, with multiple tumors or experiencing early recurrence. |
| 18 | 2018 | Kim | South Korea | 64 | | 89.00% | CIS | Treatment with at least six cycles of Bacillus Calmette-Guérin (BCG) | | Older age was also a significant factor for influencing the RFS rate. We found that the use of anti-hypertensive medications (ACEIs/ARBs) improves RFS in patients with P-CIS after BCG therapy. |

(*Continued*)

**Table 1.** (Continued)

| | Year | First Author | Country | Sample Size | Mean Age (years) | Male | Cancer Type | Treatment Protocol | Follow-up | Conclusions |
|---|---|---|---|---|---|---|---|---|---|---|
| 19 | 2018 | Racioppi | Italy | 200 | 86 and 85 | 82.50% | High-grade NMIBC | Patients in group one received six-week lies (i.e., every two weeks) intravesical BCG installations, while patients in group two received six Weekly installations (as in standard clinical practice). Patients who responded to induction treatment underwent to at least one year of BCG maintenance therapy | Two years | A customized regimen of BCG administration is possible and safe in frail elderly patients, limiting side effects and risk of undertreatment but maintaining oncological outcomes. |
| 20 | 2019 | Calo | Italy | 123 | | | high-grade T1 | - | 65 months | Elderly patients with high-grade T1 BC are not poorer candidates for BCG treatment, as they had the similar benefit and adverse reactions than those aging ≥75 years. |
| 21 | 2019 | Carrion | Spain | 65 | 87.3 | 78.50% | NMIBC | - | | Advanced age should not be a contraindication for standard therapies in Bladder cancer. A geriatric assessment could identify patients who may benefit from adjuvant therapies after TURB. |
| 22 | 2020 | Daniels | USA | 353 BCG first induction and 116 BCG second induction | 68.74±11.01 vs. 67.31 ±10.39 | 79% vs 82.8% | High-grade NMIBC | All patients receive cystoscopy approximately 6 weeks after the end of induction therapy with BCG maintenance therapy is given to responding patients weekly for 3 weeks or months 3, 6, 12, 18, 24, 30, and 36 after the induction course / second group: subsequent 2nd 6-week induction therapy for patients who recurred or persisted | 26.28 months vs. 45.42 months | The 2nd course of BCG is efficacious with a durable HgRFS, validating the recommendations of the 2016 AUA guidelines. |
| 23 | 2020 | Krajewski | Poland | 637 | 66.5±9.3 | 83.70% | CIS | 255 patients (40%) received induction course once a week for six continuous weeks), and 382 patients (60%) received induction and any maintenance (up to 3 years, 6 + 3 schedule | 57 months | Older age was not associated with BCG immunotherapy oncological outcomes or with BCG toxicity in T1HG nonmuscle invasive bladder cancer. |
| 24 | 2021 | Matsuoka | Japan | 87 | 72.6 (50–92) | 85 (97.7%) | NMIBC | Intravesical BCG was administered once a week for 6 or 8weeks, 80mg of Tokyo 172 strain in 40mL of saline or 81mg of Connaught strain in 40mL of saline were instilled per treatment with 2h of retention time. Maintenance BCG (3 weekly instillations at 3, 6, and 12months post-treatment initiation) was considered for high-risk BC | 29.7 (2–89) | The efficacy and toxicity of intravesical BCG therapy for NMIBC patients are not associated with age. Therefore, elderly patients with high-risk NMIBC should be treated in the same manner as younger patients in clinical practice. |

**Table 2. Characteristics of the studies included in the meta-analysis.**

|  | Author | Year | Country | Sample size | Age: sample size | Follow-up time |
|---|---|---|---|---|---|---|
| 1 | Fernandez-Gomez | 2008 | Spain | 1062 | ≥70: 337 | 108m |
|  |  |  |  |  | <70: 725 |  |
| 2 | Margel | 2011 | Canada | 238 | ≥70: 127 | 38m |
|  |  |  |  |  | <70: 111 |  |
| 3 | Ajili | 2013 | Tunisia | 112 | ≥70: 43 | 30m |
|  |  |  |  |  | <70: 69 |  |
| 4 | Oddens | 2014 | Belgium | 822 | ≥70: 288 | 110m |
|  |  |  |  |  | <70: 534 |  |
| 5 | Gontero | 2015 | Italy | 2451 | ≥70: 1061 | 60m |
|  |  |  |  |  | <70: 1390 |  |
| 6 | Hurle | 2018 | Italy | 185 | ≥70: 112 | 144m |
|  |  |  |  |  | <70: 73 |  |
| 7 | Calo | 2019 | Italy | 123 | ≥75: 41 | 12m |
|  |  |  |  |  | <75: 82 |  |
| 8 | Krajewski | 2020 | Poland | 637 | ≥70: 248 | 57m |
|  |  |  |  |  | <70: 389 |  |
| 9 | Richards | 2020 | USA | 39532 | ≥70: 33011 | 52 to 67 m |
|  |  |  |  |  | <70: 6521 |  |
| 10 | Matsuoka | 2021 | Japan | 87 | ≥75: 38 | 29.7 (2–89) m |
|  |  |  |  |  | <75: 49 |  |

In 8 studies, the age of elderly patients was defined as greater than 70 years old, and in one study, ≥70 years old.

The geographical distribution of studies was demonstrated in the pie diagram as six studies were from Europe, two studies from North America (Canada and U.S), and one study from Africa (Tunisia) (Fig 2).

The results from the meta-analysis of disease-free survival (DFS), progression-free survival (PFS), and cancer-specific survival (CSS) are summarized in a forest plot (Fig 3). The pooled estimates show the ratio survival rate in older patients compared to younger ones. The overall assessment of H.R. for DFS was 1.04 (95% CI: 0.85,1.26). The overall H.R. for CSS was 1.43 (95% CI: 1.11,1.83), which shows a significant association between CSS and age. The overall

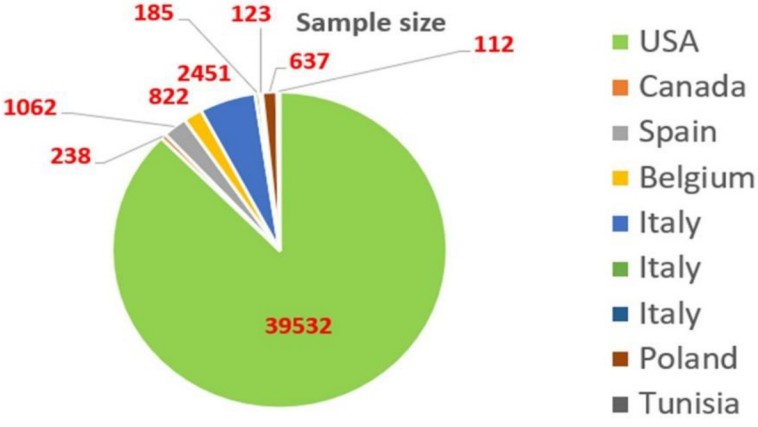

**Fig 2. The geographical distribution of studies.**

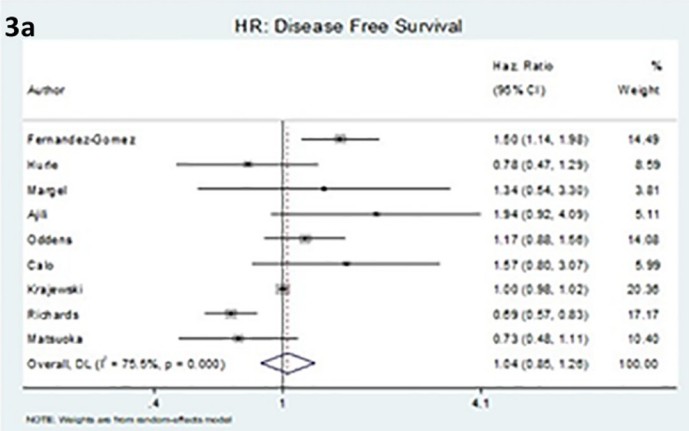

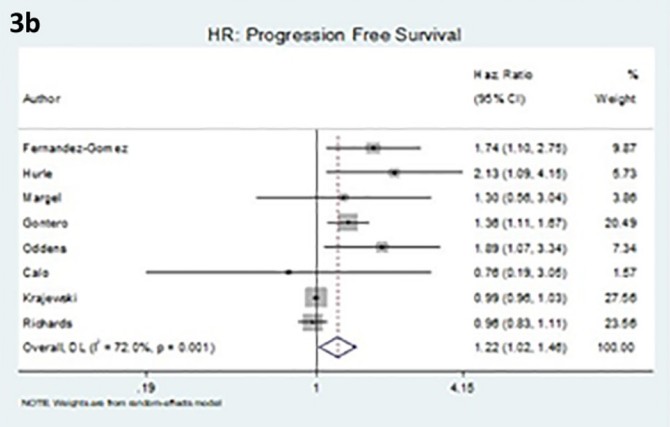

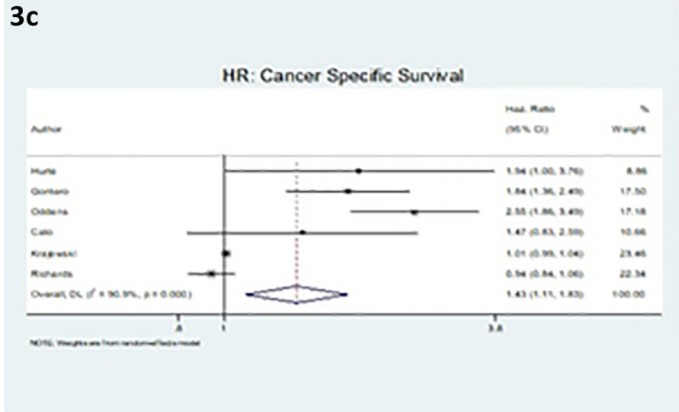

**Fig 3. Meta-analysis of the estimated hazard ratios (H.R.s) attributed to age, adjusting for other factors.** a: Overall HR of disease-free survival for eight studies (1.08). b: Overall HR of progression-free survival for 8 studies (1.22). c: Overall HR of cancer-specific survival for six studies (1.43).

estimate of H.R. for PFS showed that the survival rate was significantly higher in older age groups (H.R.: 1.22(95% CI: 1.02,1.46)). In all three forest plots, the proportion of total variation in effect estimate between studies heterogeneity (I-square) was above 50%. In order to address this, random effect analysis was used to incorporate heterogeneity among studies. The 95% prediction interval for H.R. of DFS, PFS, and CSS were (1.00, 3.49), (1.00, 3.36), and (1.00, 6.57), respectively.

The Galbraith plot analysis was used to detect sources of heterogeneity in our study. The studies of Fernande and Richards were found as outliers that were outside the confidence limits. The value of the I-square decreased to 36% (p-value = 0.153) after deleting the outliers. However, the overall estimate of H.R. for DFS remained unchanged (Fig 4).

Fig 5 shows the funnel plot to check for the existence of publication bias. According to the qualitative examination of this plot, there is evidence of negligible publication bias. However, this could not provide sufficient evidence for the number of included studies<10. Results of the Begg and Egger tests are presented in Fig 6. The statistics for Kendall τ was −0.21 (P-value = .842). In addition, the results of the Egger test showed a p-value of 0.785 (95% CI: -1.712, 2.180). The quantitative evaluation of publication bias does not suggest a considerable publication bias.

Finally, the Duval and Tweedie nonparametric "trim and fill" method was used to evaluate the publication bias further. Two additional missing studies were imputed by the trim-and-fill

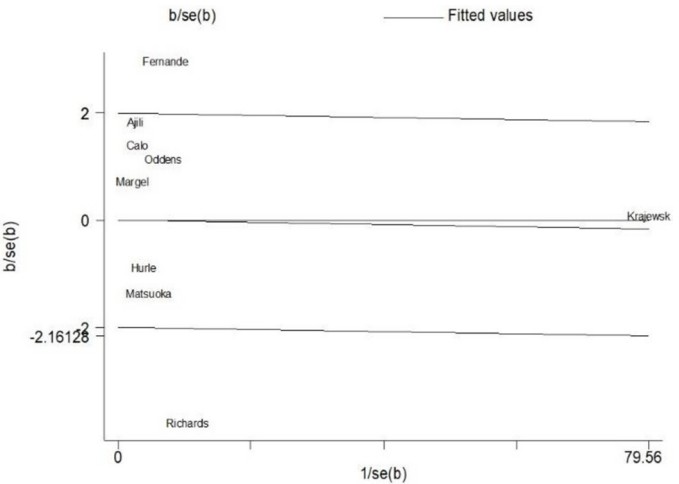

**Fig 4. Galbraith plot for indicating the sources of heterogeneity among meta-analysis results.**

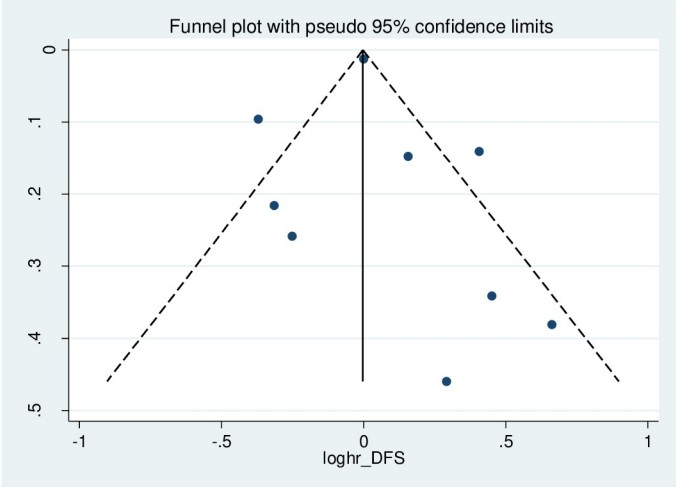

**Fig 5. The funnel plot for publication bias evaluation.**

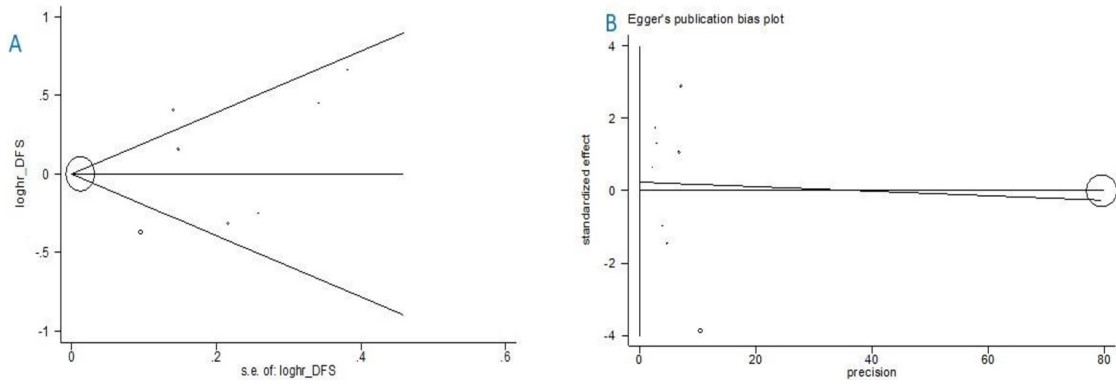

**Fig 6. (A)** Begg's funnel plot with pseudo 95% confidence limits. (**B**): Egger's publication bias plot.

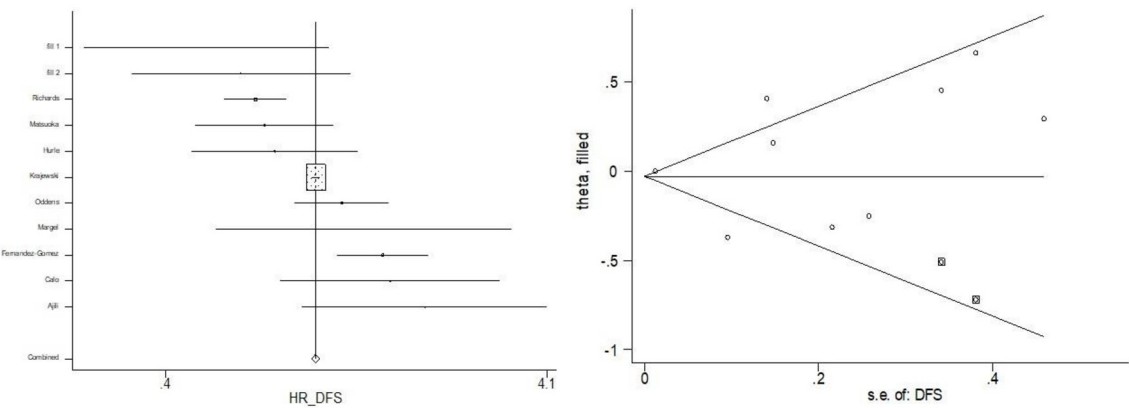

**Fig 7. Publication bias for two missing studies.**

way (Fig 7), and the overall estimate for H.R. of DFS altered from 1.04 to 0.971 (95%CI: 0.781, 1.161). However, this sensitivity testing of bias evaluation had limited power due to the low number of studies included.

## Discussion

The majority of bladder cancers are nonmuscle invasive (NMIBC) and include Ta, T1, and carcinoma in situ (CIS) that are confined to the mucosa or submucosal layers [6, 38, 39].

The standard treatment of bladder tumors is transurethral resection of tumor (TURBT) for initial pathologic diagnosis and determining the depth of tumor invasion. According to the tumor grade, size, multifocality, and recurrence, the EAU guideline developed the risk stratification for bladder cancer. The intravesical BCG instillation is recommended in intermediate (multiple and recurring Ta more than 3 cm) and high-risk patients for recurrence (T1, CIS, and high-grade tumors) as a standard protocol [6, 40]. In the united states, the preferred agent for intravesical therapy is BCG regarding the other agents [41, 42]. Nowadays, the most accepted BCG administration protocol is an initial 6-weekly intravesical instillation followed by maintenance therapy of 3-weekly installation in 3,6 and every six months interval for 1–3 years according to the risk category groups. (1 year in intermediate and 1–3 years in high-risk patients) [7]. The EAU guideline 2021 updated the individual risk stratification for BCG administration and added a very high-risk category to the risk stratification. The exciting changes are that in patients with T1 histology, the presence of multiple tumors, recurrent tumors, and age > 70 years are essential factors that could upgrade the risk stratification group from high risk to very high-risk category [6].

The most proposed mechanism of the effect of the BCG on bladder cancer is the activation of the T-cell-related immune system [43–46]. One of the continuing concerns regarding the intravesical BCG is its efficacy in elderly patients related to the probable reduced immune response with advanced age; and its safety profile[15, 23, 47]. The majority of studies included small series of patients, and more importantly, the maintenance therapy was not wholly performed. Our review included studies that NMIBC treated with TURBT and intravesical BCG installation and subsequent maintenance therapy from January 2000 to December 2020. The studies that had not completed maintenance therapy were excluded from the review. The primary endpoint of our review were oncological outcomes, including disease-free survival, progression-free survival, and cancer-specific survival. Concerning disease-free survival, eight studies meet the initial criteria for review.

In a study by Ferro et al., they evaluated the impact of age (cut-off 70 years) on predicting oncologic behavior of pure carcinoma in situ of the bladder and the response to BCG therapy in 172 patients with pure CIS. They concluded that advanced age at diagnosis appears to be associated with an increased risk of recurrence and progression of pure carcinoma in situ of the bladder after BCG therapy [48].

The Club Urologico Espanol de Tratamiento Oncologico (CUETO) group performed phase 3 studies with four prospective trials based on the different doses of intravesical BCG, with a 5–6 months maintenance schedule in all the trials and a total of 1491 cases with intermediate and high-risk tumors were enrolled in the studies; in their study patients >70 years had a lower risk of progression with maintenance BCG therapy compared to the other age groups with age less than 70 years (p-value: 0.0174) [24, 49].

In a cohort study on 238 patients, Margel et al. evaluated the effect of age on the outcome of BCG maintenance therapy in elderly patients; they compared the patients less than 75 years older than 75 years. On Multivariate analysis, age was an independent risk factor for progression with a Hazard ratio (H.R.) of 2.1(95% CI 1.7–4.9). However, maintenance therapy resulted in a significant risk of progression with an H.R. of 0.8 [15].

In a study by Ajili et al., they assessed the effect of aging on the outcome of BCG therapy in 112 patients treated with two years of maintenance therapy after TURBT. In univariate analysis, their results revealed that age has no impact on time to recurrence in patients <70 years vs.>70years (H.R.: 1.38,95%CI 0.73–2.61, p-value:0.315). Also, in multivariate analysis, the tumor size, stage, and tumor number were the only factors that affected the outcome. Still, age was not a predictor of recurrence after BCG therapy (H.R.: 1.94, 95% CI 0.92–4.09, p-value:0.82) [28].

Another study by Ferro et al. evaluated the effect of body mass index(BMI) on response to the BCG therapy following TURBT. Increasing BMI was associated with a risk of recurrence and progression(HR: 2.521 $p < 0.001$) [50].

In a comprehensive study by Odden et al., they compared the effect of BCG to epirubicin in 822 patients with intermediate and high-risk NMIBC. They treated 546 patients with BCG and 276 patients with epirubicin. With a median follow-up of 9.2 years, patients with age older than 70 years had shorter cancer-specific survival (p-value:0.049)), time to progression(p-value:0.028), and overall survival(p-value:<0.001) in the multivariate analysis compared to the patients younger than 70 years. They concluded that in patients older than 70 years, the BCG is less effective and has a worse oncological outcome, but it is still more effective than epirubicin in elderly patients [18].

In a large retrospective, multicenter study by Gontero et al. on 2451 patients with T1G3 NMIBC, they evaluated factors related to the oncological outcomes. Among them, 936 patients had received maintenance therapy. In univariate and multivariate analysis, the most critical progression factors were tumor size equal to or more than 3 cm, concomitant CIS, and age equivalent to or older than 70 years. An analysis of time to the progression and its relation to the patient's age showed there was a significant difference between patients with ages <70 years and >70 years in univariate analysis (HR:1.44,95% CI 1.20–1.73, p-value:<0.001) and multivariate In analysis (HR:1.36,95% CI 1.11–1.67, p-value:0.003) [19].

Hurle et al., in a retrospective study on 185 patients with HGT1 bladder, evaluated the effect of BCG induction and maintenance therapy on recurrence and progression-free survival (PFS). The median follow-up of patients was 93 months. In the multivariate analysis, early recurrence of the tumor (in the first three months) after TURBT (HR:3.88, p-value:<0.001), age older than 70 years (HR:2.17, p-value:0.027) and multifocality of tumors (HR:2.06, p-value:0.019) were the main predictors of the progression-free survival (PFS). They concluded

that age ≥ 70 years is a risk factor for tumor progression in HGT1 BC patients treated with BCG maintenance therapy [14].

In a prospective study on 123 patients with HGT1 BC by Calo et al., they evaluated the influence of age on oncological outcomes in patients treated with Intravesical BCG. The study encompassed 82 patients aged <75 years, and 41 patients were ≥ 75 years. The median follow-up was 65 months. In the univariable analysis, there were no significant differences between the two age categories regarding recurrent free survival (HR:1.57, p-value: 0.183) and progression-free survival (HR:0.76, p-value: 0.606), and cancer-specific survival (HR:1.47, p-value: 0.507). They concluded that patients older than 75 years had the same safety profile regarding the BCG complication and had a similar oncological outcome compared to those <75 years [32]. According to the results of this study, since the new EAU guideline categorized the elderly patients as a risk factor for disease progression, these patients should not be deprived of receiving BCG therapy [38].

Krajewski et al., in a retrospective study, analyzed the impact of age on BCG toxicity and oncological outcomes in HGT1 BC patients. Their study comprised 637 patients (389 patients <70 years and 248 ≥ 70 years) with a median follow-up of 57 months. The RFS (HR:1.00, p-value:0.615), PFS (HR:0.99, p-value:0.513), and CCS (HR:1.01, p-value:0.349) were not significantly different between the two age groups. Also, there were no significant differences between the two groups regarding moderate and severe complications (47.6% vs. 44.5%). They concluded that the elderly patients had the same oncological outcome as the younger patients with the same complication rate [35].

In an interesting large retrospective cohort study by Richards et al., they evaluated the BCG therapy pattern in NMIBC patients using the large database from the Surveillance, Epidemiology, and End Results (SEER) with a median follow-up of 52 to 67 months. The total number of patients included in the study was 39,532 patients. Among them, 18,814 (47.6%), 6280 (15.9%), 14,438(36.5%) characterized as low, intermediate, and high-risk groups respectively. The patient received adequate BCG therapy at 68.8% and 69.1% in the intermediate and high-risk groups, respectively, with an overall 53%. These results mean that about half of patients do not receive adequate treatment regarding the guidelines' clear instructions for BCG administration. The authors hypothesized that concerns related to the BCG toxicity in comorbid patients and elderly patients might explain this pattern. They concluded that adequate BCG maintenance resulted in improved oncological outcomes, especially in older patients, as their results demonstrated a significant reduction in CSS in intermediate (HR:0.72, p-value:0.015) and high risk (HR:0.79, p-value:0.002) patients compared to the low-risk groups during the first 12 months follow-up after TURBT [37].

In a recent study by Ferro et al., they evaluated the effect of the COVID-19 pandemic on the management of non-muscle invasive bladder cancer (NMIBC) on 2591 patients from 27 centers that underwent TURBT. They observed a significant delay between diagnosis and surgical treatment, with lower adherence to the standard therapeutic protocols such as BCG intravesical therapy [51].

As mentioned earlier, our review and meta-analysis encompassed ten studies that met our inclusion criteria (standard maintenance BCG therapy protocols). Of these ten studies, 9 examined H.R. for DFS, 8 examined H.R. for PFS, and 6 examined H.R. for CSS. These studies represent 45249 patients with NMIBC. For all outcomes, BCG therapy in ages higher than 70 is associated with decreased risk of recurrence, progression, and improvement in cancer-specific survival in studied patients. The overall estimate of H.R. for DFS was 1.08 (95% CI: 0.88–1.33). A significant association was observed between the overall H.R. for CSS (1.43; 95% CI: 1.11–1.83) and the overall H.R. for PFS 1.22(95% CI: 1.02–1.46). The noticeable finding of our study compared to other studies is that the BCG therapy did not significantly change the DSF.

Our study showed high heterogeneity among studies regarding some parameters, especially DFS; it may be related to some not-reported studies due to the low sample size. The other cause of this heterogeneity may relate to the low number of included articles; since we selected articles with a continuous age range and accepted protocols of intravesical BCG maintenance therapy.

The limitations regarding the studies included in the analysis are that younger patients were offered intravesical BCG treatment more often than elderly patients, because they are considered physiologically and psychologically different from younger patients.

According to these results, we should focus on the protocols to increase adherence to the guidelines instruction and not deprive elderly patients of adequate BCG maintenance therapy. The main reasons may have related to the concerns regarding BCG toxicity and its efficacy in elderly patients. Ongoing worldwide educational programs for urologists could improve practice patterns regarding BCG therapy regardless of patient's age and other comorbidities. Most of the mentioned contraindications for BCG therapy are based on small non-randomized studies.

## Conclusion

The BCG maintenance therapy improved CSS and PFS oncological outcomes in elderly patients with NMIBC. BCG therapy did not significantly change the DSF.

## Supporting information

**S1 File.**
(DOCX)

**S1 Checklist. PRISMA 2020 checklist.**
(DOCX)

**S1 Data.**
(XLSX)

## Acknowledgments

Special thanks to Sina hospital, Tehran University of Medical Sciences.

## Author Contributions

**Conceptualization:** Seyed Mohammad Kazem Aghamir.

**Data curation:** Hossein Farrokhpour, Mahin Ahmadi Pishkuhi.

**Methodology:** Fatemeh Khatami.

**Writing – original draft:** Abdolreza Mohammadi.

**Writing – review & editing:** Leonardo Oliveira Reis.

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
