## [Decision Letter · Decision Letter 0]

14 Apr 2022

PONE-D-22-07135Oncologic Outcomes of Bacillus Calmette-Guérin Therapy in Elderly Patients with Non-Muscle-Invasive Bladder Cancer: A Meta-AnalysisPLOS ONE

Dear Dr. Mohammadi,

Thank you for submitting your manuscript to PLOS ONE. After careful consideration, we feel that it has merit but does not fully meet PLOS ONE’s publication criteria as it currently stands. Therefore, we invite you to submit a revised version of the manuscript that addresses the points raised during the review process.

We look forward to receiving your revised manuscript.

Kind regards,

Giuseppe Lucarelli, M.D., Ph.D.

Academic Editor

PLOS ONE

Journal Requirements:

4. PLOS requires an ORCID iD for the corresponding author in Editorial Manager on papers submitted after December 6th, 2016. Please ensure that you have an ORCID iD and that it is validated in Editorial Manager. To do this, go to ‘Update my Information’ (in the upper left-hand corner of the main menu), and click on the Fetch/Validate link next to the ORCID field. This will take you to the ORCID site and allow you to create a new iD or authenticate a pre-existing iD in Editorial Manager. Please see the following video for instructions on linking an ORCID iD to your Editorial Manager account: https://www.youtube.com/watch?=_xcclfuvtxQ

5. Please include a separate caption for figure 7 in your manuscript.

Reviewers' comments:

Reviewer's Responses to Questions

**Comments to the Author**

1. Is the manuscript technically sound, and do the data support the conclusions?

Reviewer #1: Yes

Reviewer #2: Yes

Reviewer #3: Partly

Reviewer #4: Yes

2. Has the statistical analysis been performed appropriately and rigorously? 

Reviewer #1: Yes

Reviewer #2: Yes

Reviewer #3: Yes

Reviewer #4: Yes

3. Have the authors made all data underlying the findings in their manuscript fully available?

Reviewer #1: Yes

Reviewer #2: Yes

Reviewer #3: Yes

Reviewer #4: Yes

4. Is the manuscript presented in an intelligible fashion and written in standard English?

Reviewer #1: Yes

Reviewer #2: Yes

Reviewer #3: No

Reviewer #4: Yes

5. Review Comments to the Author

Reviewer #1: Dear authors, I carefully read your paper that I found interesting. The argument is not new but is still debated. It is still difficult finding a correct suggestion about BCG usage in elderly. I believe that the paper can be considered for publication after following some reviews. Please look following suggestions.

INTRODUCTION: first sentence, please add a reference. I believe that instead of talking about grades and Cis is more correct talking about high- and super-high risks, in accordance with latest guidelines. Please include the concept of induction and maintenance.

MATERIALS&METHODS: please enrich the section about inclusion/exclusion criteria (e.g. primary/recurrent tumors, characteristics, Cis, etc…).

DISCUSSION: the first paragraph can be moved to introduction. You can refer to EAU guidelines 2022, they have been realized. I suggest to summarize the part where your repeat results of the studies (there are tables at this regard) while I think that you should improve the discussion of the results.

Reviewer #2: Elderly Patients with Non-Muscle-Invasive Bladder Cancer: A Meta-Analysis

COMMENTS TO AUTHORS:

The aim of this manuscript is to evaluate the efficacy and toxicity of BCG therapy among

aged (>70) and younger patients with non-muscle-invasive bladder cancer (NMIBC).

The authors concluded the BCG maintenance therapy improved oncological outcomes

DFS, CSS, and PFS in elderly patients with NMIBC and had a similar safety profile without

moderate and severe complications.

I believe that the study has sufficient merit to be considered for publication on Plos One,

although major revisions are required.

MAJOR COMMENTS

- The introduction should be argued. Authors should better discuss the superiority of

BCG in preventing tumor recurrence in intermediate-risk and high-risk NMIBC

patients compared to a combination of epirubicin and interferon-alpha2b, mitomycin

C or epirubicin alone.

- Discussion. it will be for the benefit of the reader if the authors add and discuss the

role of age (cut-off 70 years) at diagnosis in predicting oncologic behavior of pure

carcinoma in situ of the bladder and the response to BCG therapy (doi:

10.1016/j.clgc.2021.12.005 ; PMID 35033480). The authors should also underline

that another limitation regarding the studies under consideration is that younger

patients were offered intravesical BCG treatment more often than elderly patients,

because they are considered physiologically and psychologically different from

younger patients. They should discuss the difficulties encountered during the Covid-

19 period as a reduced proportion of patients who underwent BCG therapy

(induction + maintenance) after surgery (doi: 10.3390/cancers13215276 ; PMID

34771440).

Reviewer #3: Objectives: Authors purposes in this manuscript were to evaluate the efficacy and toxicity of BCG therapy among aged (>70 years) and younger patients with non-muscle-invasive bladder cancer (NMIBC).

1. The second paragraph in the introduction is not intelligible, and I would suggest the authors to rephrase it.

2. If there is an acronym given to a topic I would suggest the authors to use the acronym throughout the manuscript.

3. The following paragraph: “Finally, activation of CD8+ cytotoxic, Natural Killer cells (NK), and shedding of inflammatory molecules like Interferon-gamma (IFNγ), Interleukin-12 (IL-12), and tumor necrosis factor Alfa (TNFα) (2, 3)” does not give a conclusion to the statement. It would be better if the authors will rephrase it.

4. The next phrase: “The medical reaction to intravesical BCG can be linked to the basic immune levels and activating potential which can be influenced by the age of the patient (4) is relatively poor to be understood by the English language readers. It would be better if the authors will rephrase it.

5. The purpose of this review does not provide enough information to raise the interest level for readers. In order for the readers to fully understand the main objectives of the study please provide at the end of the introduction section a more detailed presentation of the analysis that the authors are about to perform bellow.

6. The following phrase: “advanced age was defined differently from 60 to 80 in studies” is lacking probably years for 60 and 80. I would suggest the authors to specify if indeed the authors meant “years”.

7. In the headline of Table 1 on the seventh column is specified Male. Is it the percentage of male patients from each study? I would suggest the authors to clarify this headline. If the authors chose to insert a percentage it would be best to use the unit measure for all studies in the Tabel. This is also available for the sixth column (mention years for each study).

9. In Table “Treatment protocol” second row please correct “installations”

10. In the results section, it would be better for the discussions in the footnote of figures to be inserted in discussion section. It does not belong to results section.

11. The first phrase in the Discussion section should be erased as the intended information has been mentioned in the Introduction section.

12. Any country is written with capital letters. I would invite the authors to correct this.

13. The following phrase: “besides the concerns regarding the safety profile of the BCG, but the studies are conflicting results (8, 16, 45)” is not intelligible. I would suggest the authors to rephrase it.

14. In the discussion section I advise the authors to emphasize the findings of their own systematic review and meta-analysis, to compare the findings with existing relevant literature and to highlight how this study adds new value to the existing evidence. The discussion section in this form does not fulfill the main outcomes of the proposed review.

15. In the introduction and conclusions sections (abstract and manuscript) the authors mentioned the safety profile in elderly NMIBC patients treated with BCG. There is no evidence from the analyzed data from the meta-analysis about safety and toxicity in elderly patients; therefore it should be not mentioned.

Reviewer #4: This meta-analysis evaluates the efficacy and toxicity of BCG therapy among aged (>70) and younger patients with non-muscle-invasive bladder cancer. I appreciate this study and suggest to evaluate also 3 recent study in your analysis or in discussion :

-Clin Genitourin Cancer. 2022 Apr;20(2):e166-e172. doi: 10.1016/j.clgc.2021.12.005. Epub 2021 Dec 10.Clin Genitourin Cancer. 2022. PMID: 35033480

World Journal of Urology (2019) 37:507–514 https://doi.org/10.1007/s00345-018-2397-1 , see median age of this study

6. PLOS authors have the option to publish the peer review history of their article (what does this mean?). If published, this will include your full peer review and any attached files.

Reviewer #1: **Yes: **Gian Maria Busetto

Reviewer #2: No

Reviewer #3: No

Reviewer #4: No

---

## [Author Response · Author response to Decision Letter 0]

18 Apr 2022

Dear Editor 

Giuseppe Lucarelli

Thanks for the comments to change the manuscript for the better. Following is my point by point my response.

Journal Requirements:

RE: This item was checked and is correct.

RE: The authors received no specific funding for this work.

RE: Our data set is available in supplementary file 1.

4. PLOS requires an ORCID iD for the corresponding Author in Editorial Manager on papers submitted after December 6th, 2016. Please ensure that you have an ORCID iD and that it is validated in Editorial Manager. To do this, go to ‘Update my Information’ (in the upper left-hand corner of the main menu), and click on the Fetch/Validate link next to the ORCID field. This will take you to the ORCID site and allow you to create a new iD or authenticate a pre-existing iD in Editorial Manager. Please see the following video for instructions on linking an ORCID iD to your Editorial Manager account: https://www.youtube.com/watch?=_xcclfuvtxQ

RE: The ORCID iD of the corresponding Author is: 0000-0002-0483-2635

5. Please include a separate caption for figure 7 in your manuscript.

RE: This caption was added to the manuscript.

Comments to the Author

1. Is the manuscript technically sound, and do the data support the conclusions?

Reviewer #1: Yes

Reviewer #2: Yes

Reviewer #3: Partly

Reviewer #4: Yes

2. Has the statistical analysis been performed appropriately and rigorously?

Reviewer #1: Yes

Reviewer #2: Yes

Reviewer #3: Yes

Reviewer #4: Yes

3. Have the authors made all data underlying the findings in their manuscript fully available?

Reviewer #1: Yes

Reviewer #2: Yes

Reviewer #3: Yes

Reviewer #4: Yes

4. Is the manuscript presented in an intelligible fashion and written in standard English?

Reviewer #1: Yes

Reviewer #2: Yes

Reviewer #3: No

Reviewer #4: Yes

5. Review Comments to the Author

Please use the space provided to explain your answers to the questions above. You may also include additional comments for the Author, including concerns about dual publication, research ethics, or publication ethics. (Please upload your review as an attachment if it exceeds 20,000 characters)

Reviewer #1: Dear authors, I carefully read your paper that I found interesting. The argument is not new but is still debated. It is still difficult finding a correct suggestion about BCG usage in elderly. I believe that the paper can be considered for publication after following some reviews. Please look following suggestions.

INTRODUCTION: first sentence, please add a reference. I believe that instead of talking about grades and Cis is more correct talking about high- and super-high risks, in accordance with latest guidelines. Please include the concept of induction and maintenance.

RE:

-The first sentence reference was added to the manuscript

-The latest guideline reference added to the manuscript with a focus on high-grade tumor

-The concept of the induction and maintenance of BCG therapy was added to the introduction

MATERIALS&METHODS: please enrich the section about inclusion/exclusion criteria (e.g., primary/recurrent tumors, characteristics, Cis, etc.…).

RE:

Inclusion criteria were all patients with a histopathological diagnosis of non-muscle-invasive bladder cancer who received a BCG induction and maintenance therapy. We did not exclude recurrent, multiple, or CIS tumors. 

DISCUSSION: the first paragraph can be moved to introduction. You can refer to EAU guidelines 2022, they have been realized. I suggest to summarize the part where your repeat results of the studies (there are tables at this regard) while I think that you should improve the discussion of the results.

RE:

-The first paragraph moved to the introduction

-The EAU 2022 referenced and mentioned changes are made

Reviewer #2: Elderly Patients with Non-Muscle-Invasive Bladder Cancer: A Meta-Analysis

COMMENTS TO AUTHORS:

The aim of this manuscript is to evaluate the efficacy and toxicity of BCG therapy among

aged (>70) and younger patients with non-muscle-invasive bladder cancer (NMIBC).

The authors concluded the BCG maintenance therapy improved oncological outcomes

DFS, CSS, and PFS in elderly patients with NMIBC and had a similar safety profile without

moderate and severe complications.

I believe that the study has sufficient merit to be considered for publication on Plos One,

although major revisions are required.

MAJOR COMMENTS

- The introduction should be argued. Authors should better discuss the superiority of

BCG in preventing tumor recurrence in intermediate-risk and high-risk NMIBC

patients compared to a combination of epirubicin and interferon-alpha2b, mitomycin

C or epirubicin alone.

RE: superiority of BCG in preventing tumor recurrence and its reference was added to the introduction.

- Discussion. it will be for the benefit of the reader if the authors add and discuss the

role of age (cut-off 70 years) at diagnosis in predicting oncologic behavior of pure

carcinoma in situ of the bladder and the response to BCG therapy (doi:

10.1016/j.clgc.2021.12.005 ; PMID 35033480). 

RE: Thank you so much for the recommendation. The Ferro et al. study was discussed in the discussion

The authors should also underline that another limitation regarding the studies under consideration is that younger patients were offered intravesical BCG treatment more often than elderly patients,

because they are considered physiologically and psychologically different from younger patients. 

RE: Thank you so much for the comments; this item was added to the discussion

They should discuss the difficulties encountered during the Covid- 19 period as a reduced proportion of Patients who underwent BCG therapy (induction + maintenance) after surgery (doi: 0.3390/cancers13215276 ; PMID 34771440).

RE: Thank you so much for the recommendation; this study was added to the discussion

Reviewer #3: Objectives: Authors purposes in this manuscript were to evaluate the efficacy and toxicity of BCG therapy among aged (>70 years) and younger patients with non-muscle-invasive bladder cancer (NMIBC).

1. The second paragraph in the introduction is not intelligible, and I would suggest the authors to rephrase it.

RE: this paragraph was rephrased

2. If there is an acronym given to a topic I would suggest the authors to use the acronym throughout the manuscript.

RE: The acronyms applied throughout the manuscript. 

3. The following paragraph: “Finally, activation of CD8+ cytotoxic, Natural Killer cells (N.K.), and shedding of inflammatory molecules like Interferon-gamma (IFNγ), Interleukin-12 (IL-12), and tumor necrosis factor Alfa (TNFα) (2, 3)” does not give a conclusion to the statement. It would be better if the authors will rephrase it.

RE: The consequence of these inflammatory responses added to this paragraph.

4. The next phrase: “The medical reaction to intravesical BCG can be linked to the basic immune levels and activating potential which can be influenced by the age of the patient (4) is relatively poor to be understood by the English language readers. It would be better if the authors will rephrase it.

RE: Rephrased item was replaced (The therapeutic response to intravesical BCG is dependent on the response of the innate immune system, influenced by the patient's age).

5. The purpose of this review does not provide enough information to raise the interest level for readers. In order for the readers to fully understand the main objectives of the study please provide at the end of the introduction section a more detailed presentation of the analysis that the authors are about to perform bellow.

RE: this item changed, as you mentioned.

6. The following phrase: “advanced age was defined differently from 60 to 80 in studies” is lacking probably years for 60 and 80. I would suggest the authors to specify if indeed the authors meant “years”.

RE: The” years” was addad to the mentioned item.

7. In the headline of Table 1 on the seventh column is specified Male. Is it the percentage of male patients from each study? I would suggest the authors to clarify this headline. If the authors chose to insert a percentage it would be best to use the unit measure for all studies in the Tabel.

RE: The seventh column was corrected (male=63 patients (75%) ,female= 21 patients(25%)

 This is also available for the sixth column (mention years for each study).

RE: The sixth column was corrected(years).

9. In Table “Treatment protocol” second row please correct “installations”

RE: This word was corrected to the instillation

10. In the results section, it would be better for the discussions in the footnote of figures to be inserted in discussion section. It does not belong to results section.

RE: This change was made as you mentioned.

11. The first phrase in the Discussion section should be erased as the intended information has been mentioned in the Introduction section.

RE: This phrase was removed and added to the introduction.

12. Any country is written with capital letters. I would invite the authors to correct this.

RE: It seems that this item is correct in text and tables 

13. The following phrase: “besides the concerns regarding the safety profile of the BCG, but the studies are conflicting results (8, 16, 45)” is not intelligible. I would suggest the authors to rephrase it.

RE: This item was changed as you wished(One of the continuing concerns regarding the intravesical BCG is its efficacy in elderly patients related to the probable reduced immune response with advanced age; and safety profile).

14. In the discussion section I advise the authors to emphasize the findings of their own systematic review and meta-analysis, to compare the findings with existing relevant literature and to highlight how this study adds new value to the existing evidence. The discussion section in this form does not fulfill the main outcomes of the proposed review.

RE: A significant association was observed between the overall H.R. for CSS (1.43; 95% CI: 1.11-1.83) and the overall H.R. for PFS 1.22(95% CI: 1.02-1.46). The noticeable finding of our study compared to other studies is that the BCG therapy did not significantly change the DSF

15. In the introduction and conclusions sections (abstract and manuscript) the authors mentioned the safety profile in elderly NMIBC patients treated with BCG. There is no evidence from the analyzed data from the meta-analysis about safety and toxicity in elderly patients; therefore it should be not mentioned.

RE: this item was corrected in the abstract and manuscript.

Conclusion: The BCG maintenance therapy improved CSS and PFS oncological outcomes in elderly patients with NMIBC. BCG therapy did not significantly change the DSF

Reviewer #4: This meta-analysis evaluates the efficacy and toxicity of BCG therapy among aged (>70) and younger patients with non-muscle-invasive bladder cancer. I appreciate this study and suggest to evaluate also 3 recent study in your analysis or in discussion :

RE: Thank you so much for your recommendations

-Clin Genitourin Cancer. 2022 Apr;20(2):e166-e172. doi: 10.1016/j.clgc.2021.12.005. Epub 2021 Dec 10.Clin Genitourin Cancer. 2022. PMID: 35033480

RE: This articles was mentioned in the discussion

World Journal of Urology (2019) 37:507–514 https://doi.org/10.1007/s00345-018-2397-1 , see median age of this study

RE: The data of this article was added to the discussion.

---

## [Editor Report · Decision Letter 1]

20 Apr 2022

Oncologic Outcomes of Bacillus Calmette-Guérin Therapy in Elderly Patients with Non-Muscle-Invasive Bladder Cancer: A Meta-Analysis

PONE-D-22-07135R1

Dear Dr. Mohammadi,

We’re pleased to inform you that your manuscript has been judged scientifically suitable for publication and will be formally accepted for publication once it meets all outstanding technical requirements.

Kind regards,

Giuseppe Lucarelli, M.D., Ph.D.

Academic Editor

PLOS ONE
---

## [Editor Report · Acceptance letter]

28 Apr 2022

PONE-D-22-07135R1 

Oncologic Outcomes of Bacillus Calmette-Guérin Therapy in Elderly Patients with Non-Muscle-Invasive Bladder Cancer: A Meta-Analysis 

Dear Dr. Mohammadi:

I'm pleased to inform you that your manuscript has been deemed suitable for publication in PLOS ONE. Congratulations! Your manuscript is now with our production department. 

Kind regards, 

on behalf of

Dr. Giuseppe Lucarelli 

Academic Editor

PLOS ONE